# Centrality of Religiosity as a Predictor of Work Orientation Styles and Work Engagement: A Moderating Role of Gender

**Bohdan Rożnowski \*** and **Beata Zarzycka**

Institute of Psychology, The John Paul II Catholic University of Lublin, 20-950 Lublin, Poland; zarzycka@kul.pl
\* Correspondence: bohdan.roznowski@kul.pl

**Abstract:** Previous research links religiosity/spirituality with a wide variety of organizational functions and practices, and, in particular, with management processes and leadership practices. Building on Huber's concept of the centrality of religiosity, we propose that religiosity can also affect career choice and development—in particular, work orientation styles and work engagement. We also suggest that these relationships can be moderated by gender. The hypotheses were tested on a sample of 219 adult employees ($M_{age} = 37.7$, $SD_{age} = 9.2$) in a cross-sectional study. Findings provide support the religiosity–career orientation style link and the moderating function of gender in the relationship of the centrality of religiosity with work orientation styles and work engagement. Specifically, the higher the centrality of religiosity, the stronger the calling orientation among women and the higher the job orientation among men. The higher the centrality of religiosity, the stronger the vigor, dedication and absorption among women and the lower the vigor, dedication and absorption among men. Our study supports the claim that being religious is related to the acceptance of traditional worldviews on gender roles at work. However, religiosity is a source of motivation to engage at work for women, whereas for men, high religiosity can reduce engagement in work.

**Keywords:** centrality of religiosity; work orientation styles; work engagement

## 1. Introduction

Over the last few decades, there has been a growing interest in the field of religion and work (Benefiel et al. 2014). Researchers have linked religiousness or spirituality (r/s) to a wide range of organizational variables, and the research so far has confirmed the positive impact of r/s variables on work unit performance (Duchon and Plowman 2005; for a review, see Benefiel et al. 2014), organizational commitment, job involvement, organizational identification, rewards satisfaction and organizational frustration (Kolodinsky et al. 2008). The integration of religiosity and spirituality in organizational settings is currently discussed as a resource that could counterbalance contemporary problems resulting from major organizational changes, for example, downsizing, reengineering and layoffs (Benefiel et al. 2014). Although promising findings have come from Anglo-American studies (Héliot et al. 2020), research is still lacking in regards to Poland.

In this paper, we expand previous research showing that among Polish employees' religious involvement positively correlated with their attitude toward their coworkers and organization, as well as job satisfaction (Wnuk 2018). We suggest that the concept of the centrality of religiosity can help clarify the question of whether or not religiosity could be viewed as a resource for individuals' work orientation and work engagement. As Davidson and Caddell (1994) suggested that being a church member or being exposed to religious influences may not be enough to think of work as important. However, when religion is internalized, it causes some people who are already inclined to "take the

additional step of viewing it as a calling, not just a career" (p. 147). The main hypothesis of this study is that work orientation and work engagement depend on the centrality of an individual's religious construct system. This prediction is based on the insight that if the religious construct system has a central position in an individual's personal construct system, it could have an intense impact on people's behaviors. We also argue that gender could be a moderator of the relationships between centrality with a work orientation style and work engagement. In doing so, our paper provides a new insight in the discussion about the role of the centrality of religiosity in organizational settings, specifically, in work orientation and work engagement.

## 1.1. Centrality of Religiosity

Huber (2003) introduced the concept of the centrality of religiosity, which refers to the importance of religious constructs among other constructs in one's personality (Huber 2003; Kelly 1955). Centrality indicates a position of religious constructs among other constructs in personality, according to the personal construct theory (Kelly 1955). Therefore, if the religious construct system is in a central position in one's personality, it is hardly restricted by super-ordinate construct systems. Thus, it can be expected that the religious construct system should have a strong relevance for general psychological functioning and for people's experiences, interpretations and behavior (Huber and Huber 2012; Zarzycka et al. 2020). If the religious construct system is placed in a subordinate position, its functioning is restricted through many other super-ordinate construct systems. The activation of the religious construct system is under the control of alternating non-religious construct systems. If the personal religious construct system is in a marginal position in one's personality, it should be of minor importance for general psychological functioning (Huber 2004). Huber (2007) described highly religious individuals as having a central position of the religious construct system; religious individuals as having a subordinated position of the personal religious construct system; and the non-religious individuals as having a marginal position of the religious construct system. Huber (2004) and Huber and Huber (2012) confirmed that the position of religious constructs in the personal constructs system affects the political relevance of religious concepts and the experience of forgiveness by God (Huber 2004; Huber and Huber 2012). The aim of this paper was to examine how the centrality of religiosity influences an individual's work orientation and engagement at work.

## 1.2. Religiosity/Spirituality at Work

Most of the research on religiosity/spirituality has been conducted on the topic of workplace spirituality (Hill and Dik 2012). For example, Reder (1982) found that spirituality-based organizational cultures were the most productive, and through maximizing productivity, they reached dominance in the marketplace. In addition, emerging evidence has suggested that spiritually healthy workplaces have performed better (Duchon and Plowman 2005; Elm 2003; Garcia-Zamor 2003). Other scholars have suggested that employees' religious beliefs and values can also influence work-related constructs, such as work orientation and work engagement (Dik et al. 2012; Park 2012).

### 1.2.1. Work Orientation Styles

Bellah et al. (1996) proposed the job- career- and calling concept, which assumed that people differ in terms of their attitudes to work. Some are job-oriented, some are career-oriented, and some are calling-oriented. Job-oriented employees look mainly to satisfy their financial needs at work. Career-oriented employees can be characterized by looking for promotions, prestigious status, power, or accolades at work. Finally, calling-oriented employees search for meaning at work, organizing their tasks and channeling their activities in such a way that is in line with the goals of the calling that serves other people (Kasprzak 2013; Kasprzak and Michalak 2016). Work orientation style is not related to a particular profession. In each profession, one can find people showing all styles of work orientation, although it is assumed that in some professions, certain styles are more common (Wrzesniewski et al. 1997).

People desire to find work that is consistent with their religious beliefs (Lips-Wiersma 2002). The Christian tradition stresses that for religious people, issues of work attitudes may be addressed by extending a general process of discerning religious guidance, involving religious or spiritual practices (Dik et al. 2012). Therefore, the research on career choice and development has been carried out on the topic of vocational calling (Park 2012), which was derived from Weber's famous work "The Protestant Ethic and the Spirit of Capitalism". Weber (2011) formulated the question of the relationship between religion and the economy in the sense that certain types of Protestant denominations fostered the development of capitalism. One of the main factors which Weber paid attention to was the "Protestant ethic" concept of calling. Weber's ideas have been studied in various religious settings, such as Islam and Buddhism in Asia, as well as the religious roots of Indian, Jewish or Japanese societies, and the different branches of Christianity (Zabaev and Prutskova 2019). Previous research has provided some support to the argument that people are more likely to view work as a calling as their religious salience increases (Davidson and Caddell 1994; Knotts 2003). For this reason, we predicted that if the position of the religious constructs system in one's personal constructs system has a relevance for general psychological functioning (Huber and Huber 2012), it should also influence an individual's work attitude. Based on a theoretical framework and research findings, we hypothesize that:

**Hypothesis 1.** *The centrality of religiosity is positively related to a calling-oriented style and negatively related to a career-oriented style.*

### 1.2.2. Work Engagement

The concept of work engagement was introduced in 2006 by researchers from Utrecht (Schaufeli and Bakker 2010). They built a coherent model of work engagement on the basis of resource theory. Work engagement, considered as the opposite of professional burnout (Maslach and Leiter 1997), is defined in terms of energy, involvement and efficacy (Schaufeli and Bakker 2004). Thus, contrary to those who suffer from burnout, engaged employees have a sense of an energetic and effective connection with their work activities, and they see themselves as able to deal well with the demands of their job. Engagement is a positive, fulfilling, work-related state of mind that is characterized by vigor, dedication and absorption (Schaufeli and Bakker 2010). Previous research has reported that work engagement is affected by work-related factors, which included social support from colleagues and supervisors, performance feedback and employees' ability to sense or create alignment between the demands of their jobs and the resources they have to meet these demands (Bakker et al. 2008; Bakker 2011). Workers' personal attributes have less often been studied (Ilke and Warr 2011). A problem-focused coping style (Storm and Rothmann 2003), self-efficacy, organizational-based self-esteem, optimism (Xanthopoulou et al. 2007) and resilience (Bakker et al. 2011) have been confirmed as predictors of work engagement.

Values are important personal resources that have an over-arching effect across contexts. For example, a person who values work, family or religion is likely to be active (Thoresen et al. 2003) and feel good at work (Hyde and Weathington 2006). Religious involvement, as it has a relevance for general psychological functioning (Huber and Huber 2012) and operates as an ongoing motivator of perceptions, attitudes and behaviors (Park et al. 2013), can also influence employees' approach to work. Therefore, we hypothesize that:

**Hypothesis 2.** *The centrality of religiosity is positively related to work engagement.*

### 1.3. Moderating Role of Gender

Gender differentiates people both in terms of their religiousness and the way they approach work. As for religiousness, a range of studies has found that women are universally more religious than men across all societies, cultures and faiths (Pew Research Center 2016). In Western nations, women

report a greater importance of religion and spirituality in their lives than men do (Zarzycka 2011). A higher proportion of females attend church, pray daily, report religious experiences and express belief in God and life after death (Robinson et al. 2019). The Christian tradition stresses the differences between men and women and attributes them different gender roles. Becker and Hofmeister (2001) claimed that religious involvement schema, for example beliefs about gender roles, may influence men's and women's behavior in different ways. The association between religious involvement and traditionally gendered roles for men and women may also contribute to gender differences in the way individuals approach their work. Thus, religiously affiliated people are more inclined to support a worldview based on these traditional gender roles (Voicu 2009). Various studies have found that gender differences can also affect experiences within the workplace, for example, how people engage in their roles and find meaning at work (Williamson and Geldenhuys 2014). Women and men differ in their job preferences: Women prefer jobs involving personal relations and helping others, whereas men prefer jobs involving challenge and power (Konrad et al. 2000). With reference to work commitment, previous research have shown that women are more engaged and find greater fulfilment in their work than men (Williamson and Geldenhuys 2014). Men experience greater enrichment from their work than from their home, while women draw greater enrichment from family life than from their work (Rothbard 2001). Although Schaufeli and Bakker (2003) reported that men scored higher than women in dedication and absorption, Pena et al. (2012) reported that female teachers showed higher scores in all three components of work engagement, whereas Ellison and Bartkowski (2002) found that married women in conservative denominations did more hours of housework, but men did not. Thus, as gender differences affect both experiences within religiousness and the workplace, we hypothesize that they could also affect the relationships of religiousness with work orientation styles and work engagement. Therefore, we hypothesize that:

**Hypothesis 3.** *The relationship between the centrality of religiosity with work orientation styles and work engagement is moderated by gender.*

## 2. Materials and Methods

### 2.1. Participants and Procedure

The total sample comprised 219 adult employees (69% women), representing various professions, aged from 18 to 65 (M = 37.7, SD = 9.25 years). All participants were Caucasian, with Polish nationality. Most of them declared themselves as Catholics (n = 273, 95.8%). The other religious affiliations were as follows: Greek–Catholic (n = 3, 1.1%); Orthodox (n = 4, 1.4%); and Protestant (n = 5, 1.7%). The participants had higher (n = 162), secondary (n = 54) or vocational school education (n = 3). The respondents represented the following employment places, according to the Polish classification of activities: finance and insurance (16.4%); health care and social assistance (15.5%); information technology (15.5%); education (14.6%); public administration and national defense (5.9%); wholesale and retail trade (5.0%); construction (3.2%); professional, scientific and technical activities (2.3%); information and communication (1.4%); and other services (20.2%).

Data were collected from December 2018 to January 2019 through online surveys. Snowball sampling was applied. The respondents received a link to the study via e-mail, and then sent the link to their friends with a request to complete the set of questionnaires. We ensured the respondents' anonymity, and to minimize the incidence of common method bias, we divided the questionnaire into sections: Items related to different constructs were presented on separate pages.

### 2.2. Measures

We applied the centrality of religiosity scale, work orientation styles scale and Utrecht work engagement scale to our study.

### 2.2.1. Centrality of Religiosity Scale

The centrality of religiosity scale (CRS) is a measure of the importance of religious constructs in personality. It was developed by Huber (Huber and Huber 2012) and adapted into Polish by Zarzycka (2007, 2011), see also (Zarzycka et al. 2020; Zarzycka and Bartczuk 2011). The CRS consists of 15 items divided into five subscales: intellect (e.g., *How often do you think about religious issues?*); ideology (e.g., *To what extent do you believe that God or something divine exists?*); private practice (*How often do you pray?*); religious experience (*How often do you experience situations in which you have the feeling that God or something divine intervenes in your life?*); and public practice (*How often do you take part in religious services?*). The total score is a measure of the centrality of religiosity. The reliability of the CRS in the present sample was 0.97.

### 2.2.2. Occupational Orientation Styles Scale

The occupational orientation styles scale (OOSS) measures three styles of work orientation: job; career; and calling (Wrzesniewski et al. 1997). A calling-oriented style is an attitude toward work that manifests itself in a strong commitment to work for the work itself (e.g., *I feel that my work has a purpose*). A career-oriented style also manifests itself in a strong commitment, but this commitment results from the expectations of psychological rewards, such as promotion or prestige (e.g., *I would like to be promoted as soon as possible*). A job-oriented style is characterized by seeking only instrumental values, primarily economic ones, at work (e.g., *I work mainly to earn a living*) (Kasprzak 2012). The OOSS consists of 15 items, to which responses are given using a five-point Likert scale ranging from 1 (*strongly disagree*) to 5 (*strongly agree*). In the present sample, the internal consistency for the OOSS subscales amounted to 0.75 (calling), 0.74 (career) and 0.72 (job).

### 2.2.3. Utrecht Work Engagement Scale

The Utrecht work engagement scale (UWES) measures three dimensions of work engagement: vigor, dedication and absorption. Vigor refers to a high level of energy and mental resilience while working, the willingness to invest effort in one's work and persistence in the face of difficulties (e.g., *When I get up in the morning, I feel like going to work*). Dedication refers to being involved in one's work, finding meaning in one's work, being challenged and experiencing a sense of enthusiasm, inspiration and pride (e.g., *I am enthusiastic about my job*). Absorption refers to being fully concentrated and engrossed in one's work, whereby time passes quickly, and one has difficulties detaching oneself from work (e.g., *When I am working, I forget everything else around me*) (Schaufeli and Bakker 2004). We applied a nine-item scale (UWES-9) in this study. All items were scored on a seven-point frequency rating scale ranging from 0 (*never*) to 6 (*always*). In the present sample, the internal consistency for the UWES-9 subscales amounted to 0.86 (vigor), 0.83 (dedication) and 0.78 (absorption).

### 2.3. Statistical Analysis

We established whether there were correlations among the key constructs—centrality of religiosity, work orientation styles and work engagement components—by means of zero-order correlations among the CRS, OOSS and UWES subscales. In the moderation models, centrality of religiosity was included as an independent variable. Work orientation styles (job, career and calling) as well as work engagement components (vigor, dedication and absorption) were subsequently included as dependent variables. Gender was examined as a moderator of these relationships. We performed all moderation analyses using the PROCESS macro for SPSS (Hayes 2018).

## 3. Results

The descriptive statistics and correlations between all the variables included in the study are presented in Table 1. Centrality was negatively related to career-oriented style ($r = -0.25$, $p < 0.001$), but was not significantly related to calling-oriented style; thus, Hypothesis 1 was only partially

supported. None of the work engagement components (vigor, dedication and absorption) were related to centrality of religiosity; thus, Hypothesis 2 was not supported.

**Table 1.** Descriptive statistics and bivariate correlations between the variables included in the study.

|  | Centrality | Job | Career | Calling | Vigor | Dedication | Absorption |
|---|---|---|---|---|---|---|---|
| Centrality | – | | | | | | |
| Job | −0.07 | – | | | | | |
| Career | −0.25 *** | 0.02 | – | | | | |
| Calling | −0.12 | −0.45 *** | 0.17 ** | – | | | |
| Vigor | 0.05 | −0.50 *** | 0.08 | 0.64 *** | – | | |
| Dedication | 0.07 | −0.50 *** | 0.11 | 0.73 *** | 0.75 *** | – | |
| Absorption | −0.02 | −0.45 *** | 0.13 * | 0.59 *** | 0.69 *** | 0.75 *** | – |
| *M* | 3.26 | 2.81 | 3.38 | 3.31 | 3.62 | 4.28 | 3.96 |
| *SD* | 1.16 | 0.87 | 0.80 | 0.78 | 1.21 | 1.17 | 1.28 |

* $p < 0.05$; ** $p < 0.01$; *** $p < 0.001$.

To examine whether gender moderates the relationships between the centrality of religiosity with work orientation style and work engagement, two sets of moderation analyses were conducted. The variables were centered prior to the analyses (Aiken and West 1991).

In the first set of analysis, centrality was included as an independent variable in the regression models, and gender was included as a moderator. Three work orientation styles—job, career and calling—were subsequently included as dependent variables. The analysis revealed a significant interaction effect of centrality and gender to job and calling orientation styles. The interaction term accounted for a significant proportion of the variance in the job-oriented style, $\Delta R^2 = 0.07$, $\Delta F(1, 215) = 16.16$, $p < 0.001$, with over 7% of the total variance explained ($F = (3, 215) = 5.57$, $p = 0.001$). Examination of the interaction plot (Figure 1A) showed that as centrality increased in men, their job orientation increased ($b = 0.40$, $t = 3.60$, $p < 0.001$). In women, an effect of centrality on job orientation was marginally insignificant ($b = −0.16$, $t = −1.89$, $p = 0.06$). The interaction term also accounted for a significant proportion of the variance in the calling-oriented style, $\Delta R^2 = 0.02$, $\Delta F(1, 215) = 4.74$, $p = 0.031$, with 4% of the total variance explained ($F = (3, 215) = 2.96$, $p = 0.033$). Examination of the interaction plot (Figure 1B) showed a decreasing effect in men ($b = −0.21$, $t = −1.83$, $p = 0.05$): As the centrality of religious constructs increased, calling orientation decreased in men. In women, an effect of the centrality of religious constructs on calling orientation was insignificant ($b = 0.10$, $t = 1.19$, $p = 0.235$). The career-oriented style was confirmed to be a function of the centrality of the religious constructs ($b = 0.39$, $t = −1.97$, $p = 0.049$); however, gender did not moderate this relationship, $\Delta R^2 < 0.001$, $\Delta F(1, 215) = 2.23$, $p = 0.629$. Taken together, men with a high centrality of the religious construct system were more strongly job-oriented and less calling-oriented at work. There were no significant relationships between centrality of the religious construct system with job orientation and calling orientation among women. Centrality of the religious construct system decreased career orientation among both women and men.

In the second set of analysis, centrality was also included as an independent variable in the regression models and gender was included as a moderator. Three dimensions of work engagement—vigor, dedication and absorption—were subsequently included as a dependent variable in each analysis. The analysis revealed a significant interaction effect of centrality and gender to all measured components of work engagement. First, the interaction term accounted for a significant proportion of the variance in vigor, $\Delta R^2 = 0.037$, $\Delta F(1, 215) = 8.46$, $p = 0.004$, with 4% of the total variance explained ($F = (3, 215) = 2.98$, $p = 0.032$). Examination of the interaction plot (Figure 1C) showed two different effects: as centrality increased in women, their vigor also increased ($b = 0.17$, $t = 2.02$, $p = 0.044$). However, as centrality increased in men, their vigor decreased ($b = −0.24$, $t = −2.12$, $p = 0.035$). Second, the interaction term accounted for a significant proportion of the variance in dedication, $\Delta R^2 = 0.036$, $\Delta F(1, 215) = 8.10$, $p = 0.005$, with over 4% of the total variance explained ($F = (3, 215) = 4.09$, $p = 0.029$). Examination of the

interaction plot (Figure 1D) also showed two different effects of centrality on dedication: as centrality increased in women, their dedication also increased ($b = 0.18$, $t = 2.16$, $p = 0.032$). In men, an effect of centrality on dedication was negative ($b = -0.22$, $t = -1.94$, $p = 0.05$). Finally, the interaction term accounted for a significant proportion of the variance in absorption, $\Delta R^2 = 0.043$, $\Delta F(1, 215) = 9.90$, $p = 0.002$, with 6% of the total variance explained ($F = (3, 215) = 4.28$, $p = 0.006$). Examination of the interaction plot (Figure 1E) showed a decreasing effect in men: As centrality increased in men, their absorption also decreased ($b = -0.36$, $t = -3.25$, $p = 0.001$). In women, an effect of centrality on absorption was insignificant ($b = 0.08$, $t = 0.91$, $p = 0.366$). Taken together, men with a high centrality of religiosity had lower engagement at work—lower vigor, dedication and absorption—whereas women with a high centrality of religiosity had higher vigor and dedication.

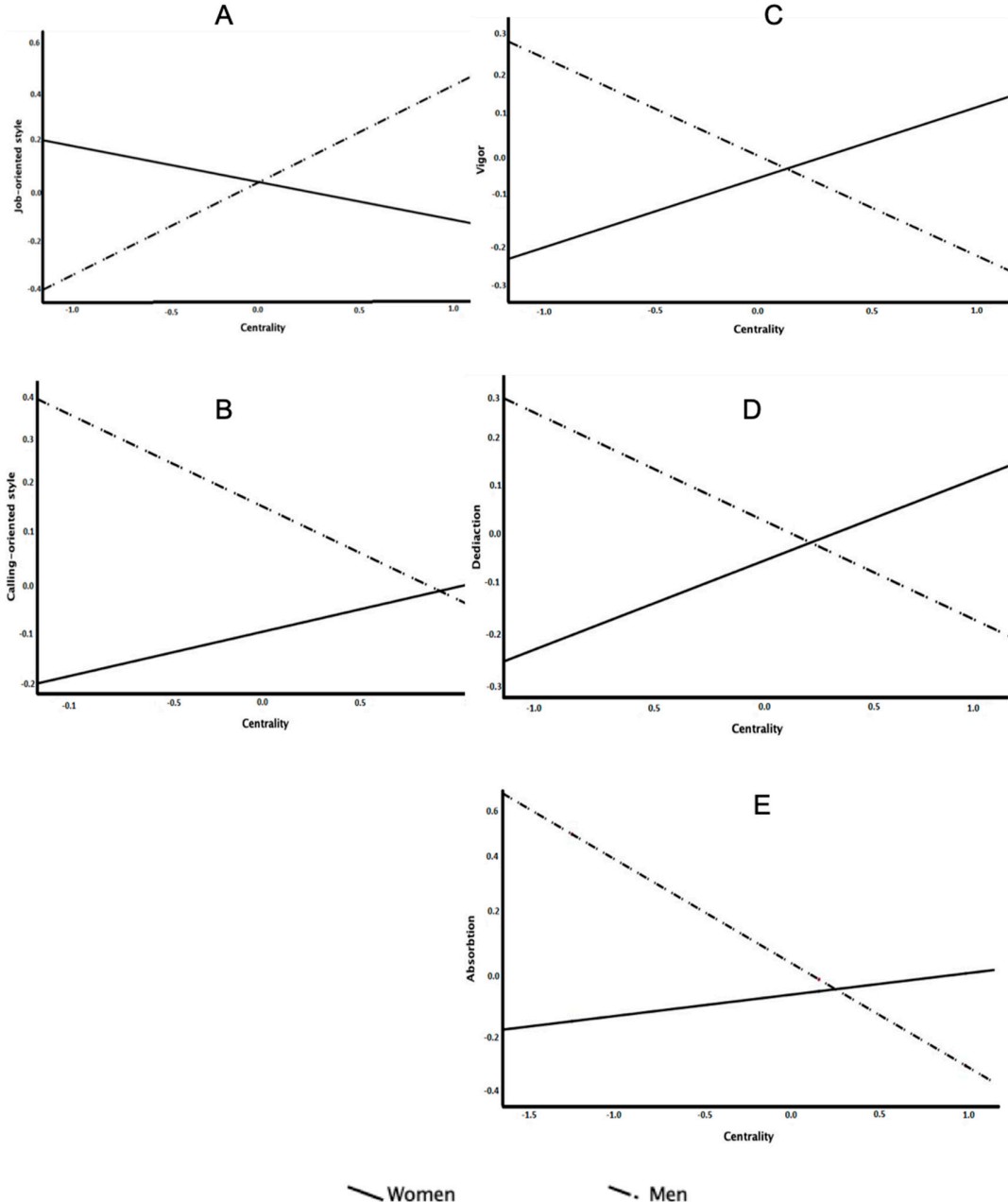

**Figure 1.** A visual representation of the moderation of the effect of the centrality of religiosity on work orientation styles: (**A**) job-orientated style and (**B**) calling-oriented style among women and men and on the work engagement dimensions: (**C**) vigor, (**D**) dedication and (**E**) absorption among women and men.

## 4. Discussion

The aim of this study was to examine the relationships among the centrality of religiosity, work orientation styles and engagement at work. We also examined whether gender moderates the relationships between the centrality of religiosity and attitudes toward work.

Partial support was found for Hypothesis 1, which stated that centrality of religiosity would be positively related to a calling-oriented style and negatively related to a career-oriented style. For the total sample, we found that the centrality of religiosity was negatively related to a career-oriented style; however, it was not related to a calling-oriented style. This result suggests that highly religious employees may be less career-oriented in comparison to fewer religious employees. However, our results, like those of Knotts (2003), did not provide support for Davidson and Davidson and Caddell's (1994) finding that religiosity fosters a "ministry" outlook on work or that people with greater religiosity tend to view work as a calling.

Hypothesis 2, which stated that centrality of religiosity would be positively related to work engagement, was not supported. For the total sample, employees with higher centrality of religiosity did not appear to engage with their jobs more than less religious employees. One explanation for the lack of association may relate to the type of respondents participating in this study. These were employees from non-religious companies. Individuals who have stronger religious commitment may have found it easier to connect their faith and work (Knotts 2003).

Our findings extend previous research focused on the relationship between religiosity and attitudes toward work, taking into account the moderating effect of gender. Hypothesis 3, which stated that relationships between the centrality of religiosity with work orientation styles and work engagement are moderated by gender, was supported. The research showed that men who are high in the centrality of religiosity tend to take a job-oriented attitude toward work, which manifests itself in seeking primarily economic values at work (Kasprzak 2012). They are also less prone to taking a calling-oriented attitude toward work, which has a more ministerial character. Furthermore, the higher the centrality of religiosity, the stronger the vigor, dedication and absorption among women and the lower the vigor, dedication and absorption among men.

Thus, our results support Dik et al. (2012) claim that for religious people, issues of work attitudes may be addressed by extending a general process of discerning religious guidance, involving religious or spiritual practices. However, this claim applies particularly to men, rather than to women. Our results may suggest that being religious may be related to acceptance of traditional worldviews on the role of men (Voicu 2009). Becker and Hofmeister (2001) reported the gendered nature of religious involvement and its relationship with work attitudes. In Becker and Hofmeister's (2001) study, men's religious involvement was associated with full-time employment, whereas women's religious involvement revealed greater complexity, intertwining with their values and lifestyles. Thus, in men, religiousness may involve adopting the traditionally defined masculine role of providing for the family (Civettini and Glass 2008).

Our results also support the claim that simply being a church member may not be enough, but when religion is internalized, it can operate as a motivator of perceptions, attitudes and behaviors across organizational contexts (Davidson and Caddell 1994; Thoresen et al. 2003; Hyde and Weathington 2006). Therefore, if religion teaches people to feel morally obligated to do good work, then religious teaching influences beliefs about appropriate attitudes, including engagement at work (Knotts 2003). However, our findings suggest that this applies to women rather than to men. Women who were high in the centrality of religiosity had higher engagement at work—higher vigor and dedication. By contrast, men high in the centrality of religiosity had lower engagement at work: lower vigor, dedication and absorption. A negative relationship between the centrality of religiosity and work engagement for men was not anticipated. However, an explanation for gender difference in the correlations between religiosity variables and work engagement may be based on group membership. Knotts (2003) observed opposite patterns of correlations between religiosity and job involvement in Protestant and non-Protestant groups. Explaining this, Knotts (2003) referred

to the more highly ascetic lifestyle and higher work ethic among Protestant than non-Protestant denominations. Thus, when comparing women's and men's participation in religion, women appear to be more religious than men (Ozorak 1996). Women are more likely than men to be church-affiliated, to feel close to God and to pray frequently (Zarzycka 2007, 2009). Consequently, religion may be a stronger motivator of work engagement among women than men. Furthermore, religious men consider their work as a way to make money, whereas religious women may be more prone to consider it as a calling. This could explain why there is not a direct influence of religiousness on engagement at work in men. This explanation may be supported by Civettini and Glass' (2008) findings, which suggest that religious conservatism does not hasten the transition to adulthood among men and is associated with lower wages.

As with all research, the present study needs to be interpreted in the context of its limitations. First of all, our study had a correlational design, which limits our ability to draw causal conclusions about the findings. Second, we used only self-report instruments to measure the variables and response bias could not be controlled. Finally, the respondents were a convenience sample of adults, dominated by women (only 31 per cent of subjects were men) and Polish Roman Catholics. Therefore, we cannot generalize our findings to all Polish adults or to representatives of other nationalities or religious groups. Furthermore, we would like to point out that our study did not collect any religious contents and put them in relation to centrality. It seems that the relevance of the religious concept of calling (Weber's Protestantism thesis) for the spirituality of the respondents may play a role. This should be investigated in future studies.

## 5. Conclusions

The current study shows that religiosity affects both work orientation styles and work engagement. However, these relationships vary according to gender. Our study supports the claim that being religious is related to acceptance of the traditional worldview on gender roles at work (Voicu 2009). However, religiosity is a source of motivation to engage at work for women, whereas for men, high religiosity can reduce work engagement. Our study also shows the relevance of the centrality of religiosity concept as a useful notion for the study of religious beliefs and values in work-related context. Additional research is needed to test the basic premise that men and women differently relate religiosity to their work orientation and work engagement.

**Author Contributions:** Conceptualization, B.Z. and B.R.; data curation, B.R.; formal analysis, B.Z.; funding acquisition, B.R.; investigation, B.R.; methodology, B.Z.; project administration, B.R. and B.Z.; writing—original draft, B.R. and B.Z.; writing—review and editing, B.R. and B.Z. All authors have read and agreed to the published version of the manuscript.

**Funding:** The APC was funded by The John Paul II Catholic University of Lublin.

**Acknowledgments:** Special thanks go to Dawid Zawalisz for his assistance in the research.

**Conflicts of Interest:** The authors declare no conflicts of interest. The funding sponsors had no role in the design of the study; in the collection, analyses or interpretation of data; in the writing of the manuscript and in the decision to publish the results.

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
