# Peer review of "Centrality of Religiosity as a Predictor of Work Orientation Styles and Work Engagement: A Moderating Role of Gender"

_religions, doi:10.3390/rel11080387_

Round 1
Reviewer 1 Report
Feedback to Author(s)
This is an interesting topic that is clearly a strong fit for reading of the journal. However, the conceptual background and logic for the variables studied is unclear which together with the somewhat weak findings limites the impact of the paper in its current form. I have outlined several points that should be addressed if the author(s) elect to complete a revision of the paper.
1. First and most importantly, the overall framing of the paper lacks clarity and consistency. The title focuses on occupational orientation, yet the review of literature discussed job orientation. Are these the same thing? A rationale for why this work is important is that these constructs are not typically studied on a sample from Poland, yet the paper includes no measure related to culture and the rationale for why this national sample is linked to the variables studied is missing.
2. More detail within the introduction on the notion of “personal construct” system is needed. This appears to be a central concept within the paper, yet it is absent within the methodology and results. In addition, the review of previous research on spirituality at work is inadequate.
3. The rationale for why gender is included as a variable and why its is included as a moderator needs more information and rationale. In addition, the gender composition of the sample is not provided. Also, looking at gender as a moderator versus examining separate regressions for men versus women is curious if the idea being tested is that the process for these variables is distinctly different for me versus women.
4. The focal outcome variable needs to be made clearer early within the introduction. It appears to change from occupational orientation to work engagement and at one point “career choice” is mentioned. This is confusing to the reader and makes the presentation of results look more like an exploratory expedition than a test of specific hypotheses.
5. A significant amount of the introduction is focused on the concept of “centrality”, yet it is not clearly measured within the study. More thought on how centrality could be measured given the existing data is critical. In addition, there is an assumption that the “shifting” of one’s central views is a deliberate and conscious process yet no research is cited to support this assumption.
6. The author(s) mention the existing research on “calling” and seem to make a case that it lacks a “religious framework”. This is an interesting and unique perspective that perhaps can somehow be validated in their date, yet this notion is not carried through to the analysis, results and discussion.
7. A key issues that the authors must consider is whether or not a focus on the notion of “calling” and how to connect calling to centrality of religiosity and a predictor of work engagement would be important in any revision of this paper. Since measures of religiosity and calling are collected, the authors might consider some type of clustering analysis (e.g., k-means) or grouping technique that would identify those with high versus low religious centrality with job/career versus calling and examine difference in level of work engagement. There are other approached to get at this relationships, but this is one example that would provide a clearer contribution to the existing research and better situation the paper within the calling literature which would be a stronger positioning that what is currently described.
8. The findings are very weak and only the gender as moderator comes out as a strong outcome. Perhaps making gender a more focal point of the paper would make sense given these findings. In addition, the readability of the graphs within Figure 1 is extremely low and does not add anything substantive within the results section.
Author Response
The following changes regarded to reviewer 1 comments have been made in accordance with the Guest Editor’s suggestions:
- Key criticisms of R1 relate to the concept of "personal construct" and the validity of the "Centrality of Religiosity Scale (CRS)". These criticisms are not valid in my opinion. The concept "personal construct" is sufficiently introduced by the authors. It serves essentially to explain the logic of the CRS and has succeeded in this respect. Furthermore, the validity of CRSs has been proven in hundreds of studies in different religious and cultural contexts. It is precisely for this reason that a "special issue" on "Research with the Centrality of Religiosity Scale (CRS)" appears in RELIGIONS - see: https://www.mdpi.com/journal/religions/special_issues/CRS In this context the article fits well.
- Thank you for this comment. Following the Guest Editor’s recommendation we did not introduce changes into sections referring to the concept "personal construct" and the CRS.
- The other criticisms of R1 can be well taken into account in the revision of the article, which I suggest to the authors. This refers in particular to the term "job orientation" instead of "occupational orientation" and a stronger reflection of gender as a variable.
- Thank you for this comment. We agree that “occupational orientation” is not the best term and that there are a few of terms in the article that may cause confusion. We decided to introduce the term “work orientation” instead of "occupational orientation" because “job orientation” is one of the subscales of the Occupational Orientation Styles Scale. So, in order to differentiate between a subscale and the general construct consisting of three subscales—job, calling, and career—we named the construct “work orientation”. The above change was introduced throughout the whole article. We also made further extended reflection on gender as a variable.
- They should also look more closely at "calling" in the context of a "religious framework". I recommend as a literature reference: Zabaev & Prutskova (2019) The Calling and Humility Scale: Extending the Weberian Approach to the Research of the Elective Affinity between Religion and the Economy - DOI: 10.17323/1728-192x-2019-2-62-88 3)
- Thank you for this comment. We looked at “calling” in the context of a “religious framework” more closely. We made the reference to the article the Guest Editor
- The cluster analyses and separate analyses of female (69%) and male (31%) respondents proposed by R1 were not necessary, as the interaction analyses are more far-reaching.
- Thank you for this comment. Yes, we agree with the Guest Editor.
Reviewer 2 Report
I don't really have any specific suggestions for improvement, other than a serious revision of the English language.
Author Response
Response to Reviewer 2 Comments
R2 criticizes above all: "I do have the fundamental question whether this research does contribute extra to already existing insights, this is very limited in scope and cannot be generalized. So the question is more to the publisher, because the scientific added value is very low. He returns the question to the publisher. As Guest Editor, I can say the following about this: The article is not only in the context of the general discussion of the connection between religiosity, job orientation and work engagement, but also in the context of the religious-psychological discussion about the concept of centrality and the fertility of this concept in various fields of research. The paper is relevant for this specific context. This context should be deepened by the authors in the discussion. They could, for example, take up the following statement by Davidson & Caddell (1994): "Our findings suggest that simply being a church member and being exposed to religious influences is not enough (denominational affiliation, pastoral influences, and sermons had no effect). But when religion is internalized, it causes some people who are already inclined to think of work as important to take the additional step of viewing it as a calling, not just a career" (p. 147). Furthermore, they could point out that the study did not collect any religious contents and put them in relation to centrality. It seems relevant to me, for example, that the relevance of the religious concept of calling (Weber's Protestantism thesis) for the spirituality of the respondents plays a role. This should be investigated in future studies.
- Thank you for this comment. We added the sentence the Guest Editor suggested into the Discussion section.
Reviewer 3 Report
In the sentences located around lines 36-38 there seems to be some words missing, including a verb. It is as if this sentence needs a subordinate sentence. Perhaps something was accidentally deleted?
Author Response
Response to Reviewer 3 Comments
- Thank you for your positive opinion.
- The article was proofread by the professional proofreading service.
Round 2
Reviewer 1 Report
The authors provided limited response to my previous points #1, 2 and 3 and insufficient or no response to previous points #4,5,6 and 8.